# Biochar reduces early-stage mineralization rates of plant residues more in coarse than fine-texture soils – an artificial soil approach.

Thiago M. Inagaki [1], Simon Weldon [1], Franziska B. Bucka [2,3], Eva Farkas [1], Daniel P. Rasse [1].

[1] Norwegian Institute of Bioeconomy Research (NIBIO). Department of Biogeochemistry and Soil Quality. Høgskoleveien 7, 1430 Ås, Norge

[2] Technical University of Munich. TUM School of Life Sciences. Chair of Soil Science. Emil-Ramann-Straße 2, 85354, Freising, Germany

[3] Goethe University Frankfurt. Institute of Physical Geography. Altenhöferallee 1, 60438 Frankfurt am Main, Germany.

*Correspondence to*: Thiago M. Inagaki (thiago.inagaki@nibio.no), Franziska B. Bucka (bucka@em.uni-frankfurt.de)

**Abstract.** Quantifying the impact of biochar on carbon persistence across soil textures is complex, owing to the variability in soil conditions. Using artificial soils with precise textural and mineral composition, we could disentangle the effects of biochar from the effects of soil particle size. We can show that biochar application significantly reduces early-stage carbon mineralization rates of plant residues in various soil textures (from 5 to 41% clay) but more significantly in sandy soils. Clay and silt particles alone also reduced C mineralization, but the magnitude of the changes was negligible compared to the impact of biochar. This finding suggests that biochar can compensate for the lack of clay in promoting C persistence in soil systems. This short report substantially contributes to understanding soil texture and biochar application interactions.

**1 Introduction**

Biochar application in agriculture has been recognized to enhance carbon sequestration and improve soil quality (Lehmann, 2007). In this regard, soil texture may play a fundamental role in the overall effectiveness of biochar in promoting carbon persistence, mainly due to influences on soil structural properties (Wang et al., 2017). However, the mechanisms behind the interaction between biochar and soil texture in promoting C storage still need to be fully understood due to the high heterogeneity in soil properties and climate conditions of natural soils. As a result, the findings in the literature on this topic can vary depending on the specific experimental conditions used. For example, Gross et al. (2021) observed higher increases in SOC stocks in more clay soils than sandy ones when evaluating field experiments in a meta-analysis. Contrastingly, the

authors observed the opposite trend when considering non-field experiments. The reasons behind it were attributed to initial
soil C contents, as low C soils have a higher potential for promoting increases in SOC stocks. However, the lack of research
examining the effects of biochar on soil organic carbon (SOC) storage under diverse soil types and conditions undermines our
understanding of these interactions. Likewise, biochar is recognized for promoting a liming effect and ameliorating acidic soils
(Bolan et al., 2023). In contrasting comparisons, such liming effects are observed more intensively in clay soils than in sandy
ones (Ajayi and Rainer, 2017), suggesting a potential synergistic effect of clay and biochar in promoting increases in soil pH.
Nonetheless, there is a lack of studies systematically investigating the relationship between biochar-induced pH changes across
a gradient of soil textures. High heterogeneity of soil properties across textural gradients usually challenges interpretation
about the specific influence of soil particle size in these interactions.
In this sense, using artificial soil with known particle size and mineral composition provides an excellent base for
understanding the mechanisms behind soil processes (Pronk et al., 2012). These artificial soils can be mixed to mimic the
composition of typical arable soils of the temperate region, while their individual properties, like soil texture, can be freely
adjusted (Bucka et al., 2021a). The use of artificial soils was already in research with mechanistic goals, such as studying the
effects of microbial activity and mineral interactions on aggregation (Pronk et al., 2012; Vogel et al., 2014; Bucka et al., 2019),
and more applied uses, such as evaluating early soil development in post-mining soils (Bucka et al., 2021b). Therefore, this
experimental setup provides a unique opportunity to investigate the intricate relationships between biochar application and soil
particle size distribution, an area that remains largely unexplored.
This short communication explored the interactions between soil texture and biochar application in the early-stage
soil organic matter mineralization using artificial soils with precise mineral and textural composition in a controlled
microcosm. We hypothesized that biochar could reduce organic soil mineralization, especially in coarser textured soils.
**2. Material and methods**
**2.1 Artificial soil preparation: texture range**
To produce the different textures of artificial soils, we have used quartz grains of varying particle sizes (Euroquarz,
Laußnitz, Germany, and Quarzwerke, Frechen, Germany). We added goethite (<6 µm), illite (<7 µm), and bentonite (<63 µm)
(Aspanger, Aspang, Austria) to create soil-like reactive surfaces in the fine silt and clay-size fraction. The C-content of the
artificial soils was considered negligible due to the insignificant C concentration of the ingredients. The soil mixtures were
prepared according to the proportions presented in Table 1. The texture classes were defined as 1) Loamy Sand, 2) Sandy
Loam, 3) Loam, 4) Clay Loam, and 5) Silty Clay.

**Table 1:** Composition of the artificial soils.

| Fraction | Ingredients (median grain size) | Soil 1 Loamy Sand | Soil 2 Sandy Loam | Soil 3 Loam | Soil 4 Clay Loam | Soil 5 Silty Clay |
|---|---|---|---|---|---|---|
| | | ------------------ Proportion (mass %) ------------------ | | | | |
| Sand | Quartz Sand (75 µm) | 81 | 61 | 45 | 21 | 8 |
| Silt | Quartz Silt (26 µm) | 14 | 29 | 39 | 48 | 51 |
| Clay | Quartz Clay (5.2 µm) | 4.45 | 8.9 | 14.24 | 27.59 | 36.49 |
| | Goethite ($< 6.3$ µm) | 0.05 | 0.1 | 0.16 | 0.31 | 0.41 |
| | Illite (MICA SFG 75) (4 µm) | 0.25 | 0.5 | 0.8 | 1.55 | 2.05 |
| | Bentonite ($< 63$ µm) | 0.25 | 0.5 | 0.8 | 1.55 | 2.05 |
| | Total | 100 | 100 | 100 | 100 | 100 |


The artificial soils were prepared by mixing the ingredients in a dry state, ranging from coarser to finer-scale particles.
As each component was added, 10 manual turns were executed to combine them. After all the ingredients were mixed, the
containers were added to a horizontal shaker and shaken overnight at 140 rpm. After the overnight shaking period, each soil
mix container was manually turned 30 times.
**2.2 Incubation Experiment and Treatments**
The experimental setup was a 5 x 2 factorial design testing 5 different textures of artificial soils with and without
biochar application with three replicates. For the incubation experiment, we used 20 g of artificial soil per experimental unit
in 120 ml glass flasks. We added ball-milled air-dried clover biomass (*Trifolium sp.*) (C:N ratio of 18) as an organic matter
source to all samples at a rate of 27 mg C g$^{-1}$ soil to mimic a natural background OC content of arable topsoils. Ball-milled
organic matter was chosen to enhance the development of these soils since the inorganic components added are C-free. We
have used dissolved organic matter extracted from a local crop field as an inoculum of soil microorganisms to the artificial
soils at a rate of 0.06 ml g soil$^{-1}$, according to Pronk et al. (2012). The artificial soils were incubated under 60% of the maximum
water hold capacity to ensure microbial activity (Supplementary Figure 1). The water used to add the inoculum was accounted
for in the amount added. Biochar was produced from Norwegian Spruce at $700^{\circ}C$, with 7 minutes holding time, and added to
the soil at a rate of 50 mg of biochar $g^{-1}$ soil. The biochar had a C content of 95.6%. The added biochar had a size distribution
between 0.063 and 2 mm, controlled by sieving before addition in the soil. The basic biochar properties are described in the
Supplementary Table 1.
**2.3 $CO_2$ respiration measurements**
We measured $CO_2$ production over 115 hours using an automated incubation system described in detail by Molstad
et al. (2007) with modifications in Molstad et al. (2016). The system consists of an autosampler (CTC PAL) connected to an
Agilent gas Chromatograph (Model 7890A, Agilent, Santa Clara, CA, USA). The system allows for high-resolution analysis
of headspace gas concentrations in airtight 120 ml serum bottles. Corrections were applied to adjust for sampling dilution,
leakage, and $CO_2$ equilibrium state as a function of the material pH and soil solution volume (Appelo and Postma, 2005). We
monitored the weight of the samples during the incubation period to check whether adjusting the water content was necessary.
Since this was a relatively short incubation (5 days) conducted in a closed environment, the water loss was minimal, and we
did not need to refill the flasks to maintain the initial moisture content.

**2.4 Soil pH analysis**
Soil pH was measured in a 0.01 M $CaCl_2$ solution according to the ISO 10390:2021.

**2.5 Statistical analysis**
We performed the Shapiro-Wilk test to confirm the data's normal distribution. Once the data normality was confirmed,
we conducted a two-way ANOVA and a posthoc test using Fisher's Least Significant Difference test (LSD) to compare biochar
and control treatments within each soil textural group. We have used linear regressions correlating the increase of clay and silt
content with the aimed variables to compare the different textural classes.

**3. Results**

## 3.1 Early plant mineralization affected by particle size and biochar applications


We observed significant effects of both biochar application and texture on C mineralization and a significant
interaction between these variables in the analysis of variance (ANOVA) at $p < 0.01$. Biochar application significantly reduced
C-mineralization rates compared to control in the coarser-textured soils: 1) Loamy Sand, 2) Sandy Loam, and 3) Loam (Figure
1) according to the LSD test at $p < 0.05$. However, no differences were observed between biochar and control treatments for
the finer-textured soils: 4) Clay Loam and 5) Silty Clay (Figure 1).

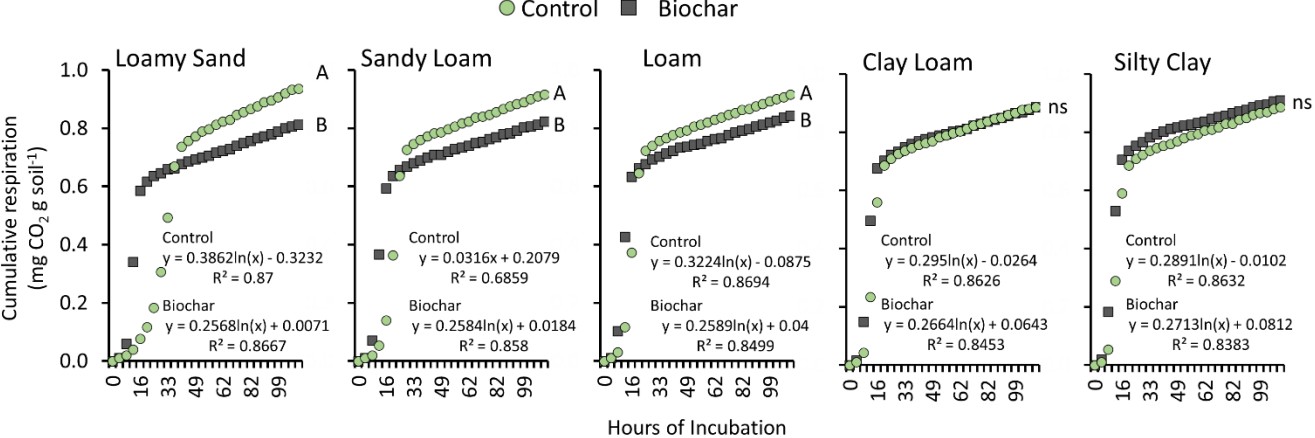


**Figure 1:** Biochar effect on the $CO_2$ cumulative respiration in artificial soils with different textures in a 5-day incubation
experiment. For a given soil, different uppercase letters indicate a significant difference ($p < 0.05$) in the cumulative respiration
between control and biochar-amended soils by Fisher's Least Significant Difference test (LSD). Each point is the average of
three replicates. Variations between replicates are not visible in the graph scale due to the uniformity of the artificial soil
composition.

The reduction in C mineralization promoted by biochar in the Loamy sand (Soil 1) was over 6-fold higher than in the
Silt Clay soil (Soil 5). The influence of clay and silt content on C mineralization also differed depending on whether biochar
was applied or not (Figure 2). However, the impact of soil texture was minimal in magnitude compared to the changes
promoted by biochar addition. The control samples' clay and silt content generally decreased C mineralization (Figure 2).
Every mg of silt and clay-sized particles in the artificial soils reduced clover residue mineralization by 0.00007 mg $g^{-1}$ soil
(Figure 2) under a constant moisture level (60% of the water hold capacity), given the slope of the equation. After a 5-day
incubation in the Silt Clay soil (Soil 5), the clover mineralization was 0.06 mg $CO_2$ $g^{-1}$ soil lower than in the Loamy Sand soil
(Soil 1).

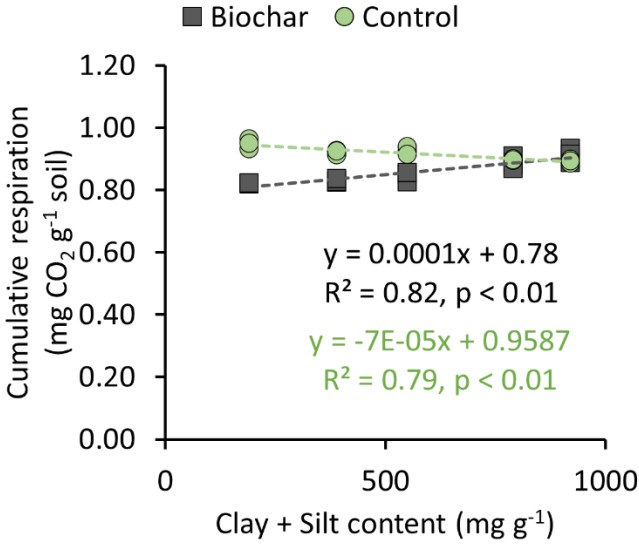


**Figure 2:** Influence of clay and silt fraction from artificial soils on the cumulative respiration in soils with and without biochar
in a 5-day incubation experiment. ** $p < 0.01$.

**3.2 Soil pH affected by particle size and biochar application**

123        Soil pH ($CaCl_2$) was significantly influenced by increased soil texture and biochar application (Figure 3). Increases

in pH due to biochar addition were overall higher with increasing clay and silt contents, i.e., by 0.68 in the Silt Clay and 0.24
in the Loamy Sand (Figure 3a). The increase of silt and clay particles alone also promoted increases in soil pH ($p<0.05$) (Figure
3b). Given the slope of the linear regression equation, every mg of clay and silt fraction promoted an increase of 0.0003 in soil
pH (CaCl2). This increase was three times higher (0.009) when biochar was applied ($p<0.01$) (Figure 2b).

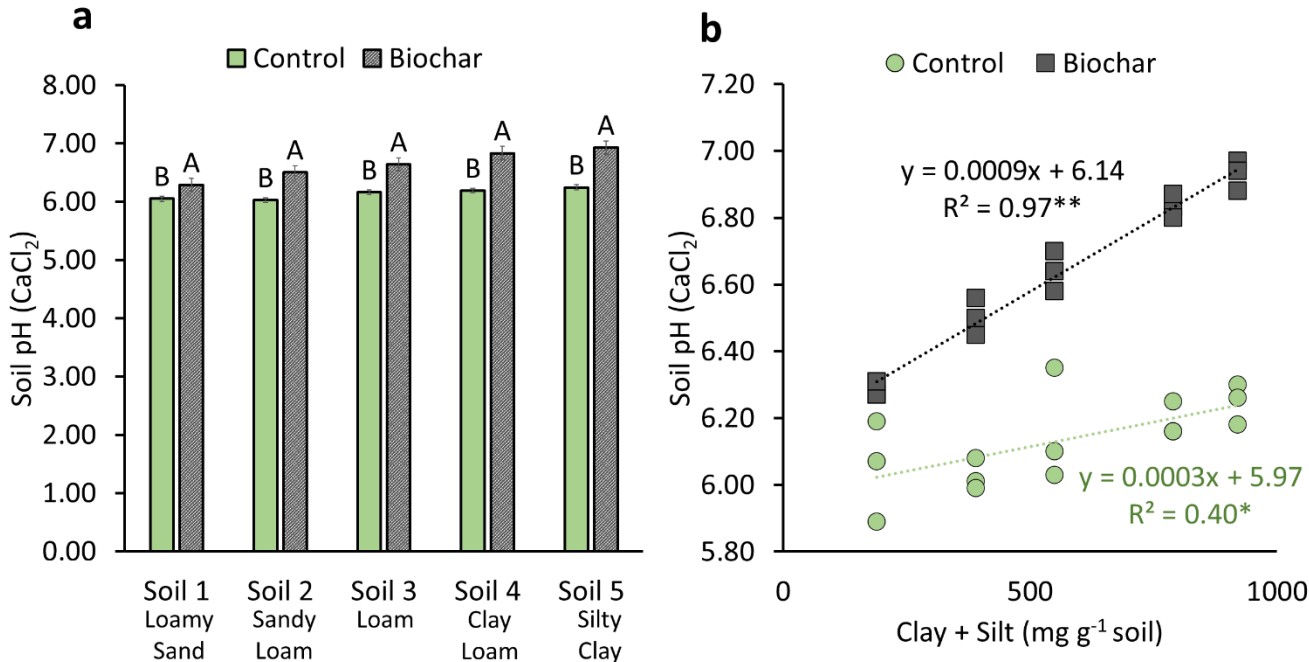

**Figure 3:** Soil pH as affected by the biochar application and soil texture in artificial soils incubated for 5 days. a) Influence of biochar application on soil pH ($CaCl_2$) under different textures. For a given soil, different letters indicate significant differences by the LSD test at $p < 0.05$. b) Influence of soil texture (clay+silt proportion) on soil pH (CaCl2) in soils with and without biochar. ** Significant at $p < 0.01$. * Significant at $p < 0.05$

## 4. Discussion

### 4.1 Interactions of clay + silt content and biochar application in reducing early mineralization of plant residues.

The significant interaction between biochar application and soil texture suggests that the extent of biochar's impact on the mineralization of plant residues depends on soil texture, as observed by different studies (Butnan et al., 2017; Gross et al., 2021; Wang et al., 2017). On the other hand, soil texture alone had minor effects on C mineralization. The differences in the magnitude of early plant mineralization depending on soil texture suggest that biochar had a higher impact on reducing early C-mineralization from clover residues in sandier textures than in clay-rich soils, which has also been observed under

natural soil conditions (Butnan et al., 2017). Biochar has known benefits in enhancing soil properties associated with promoting
C protection, such as aggregation (Wang et al., 2017; Juriga et al., 2021), sorption capacity (Siedt et al., 2021), water retention,
and porosity (Obia et al., 2016) in natural soils. Such effects may have potentially influenced the decrease of early
mineralization of the added plant residues since the improvement of soil structure in artificial soils, such as aggregate
formation, is already observed within the first days of their development (Pronk et al., 2012).
Improvements in soil structure and C stability upon biochar application are known to happen at a higher intensity in
sandy soils compared to more clay ones (Buchkina et al., 2017; Butnan et al., 2017), which may explain the biochar's better
performance in reducing early plant mineralization in sandy soils in our study. Our observations corroborate with general
trends observed in non-field experiments (Gross et al., 2021), with higher biochar performance in sandy textures compared to
clay. On the other hand, other experiments in incubation setups, such as Wang et al. (2017) and the field experiment
observations made by Gross et al. (2021), suggest an opposite trend. The reasons for these discrepancies seem to depend mainly
on the initial condition of the evaluated soils. For example, in the incubation setup comparing contrasting textures in Wang et
al. (2017), the unamended samples had no difference in water-stable aggregates despite their contrasting texture, and the C
content was higher in the sandy soil. This experiment resulted in higher impacts of biochar application on clay soils' structure
and C storage. On the other hand, in our experiment, the unamended samples with higher clay content had a significantly
higher water-hold capacity (Supplementary Figure 1), indicating an overall better structure than the sandy soils, and the organic
matter added to them was the same for all soils, which resulted in a better performance of biochar in more sandy textures.
Therefore, the impacts of biochar on reducing C mineralization in our experiment suggest that the amendment can compensate
for the lack of soil structural quality in promoting early C persistence in soils.
Likewise, Gross et al. (2021) pointed out the low initial soil C content of sandier textures as a reason for observing
higher SOC sequestration potential in non-field experiments. The contrast with field experiments can also be due to factors
that happen only in field trial setups, such as biochar leaching in sandy soils, which are not observed under pot or incubation
setups. Therefore, our interpretations are focused more on the intrinsic relationships between soil texture and biochar and their
impacts on C mineralization in a controlled environment. However, climatic factors and long-term effects need to be confirmed
through long-term field trials. Nonetheless, since most of the works evaluating the performance of biochar in different soil

textures vary in other factors such as C content, mineral composition, and structural properties, our work offers important insights to understanding these interactions of biochar with soil particle size in C mineralization.

Also, the increase of fine particles in soils is assumed to decrease organic matter decomposability due to the increased opportunity for physicochemical protection (Hassink, 1992; Kravchenko and Guber, 2017), which may explain the minor reductions observed in the mineralization rates caused by increased silt and clay fractions (Figure 2). However, despite the significant correlations, the magnitude of changes in mineralization caused by soil texture was negligible in this experiment, given the low slope values (Figure 2). Therefore, our results suggest that soil particle size (i.e., texture) has played a milder overall role than the application of biochar in our short-term study.

Especially in sand-rich soil, where the available mineral surface area (as well as the permanently charged surfaces of clay minerals) is low, biochar can deliver additional surface area for adsorption processes. The biochar effect is probably reduced by increasing clay content as the higher clay mineral surface area, and the smaller pores may overrule the biochar effect on the physicochemical protection of OM. Since the biochar added in our experiment was a standard size between 0.063 and 2 mm, we can argue that the biochar composition was foremost responsible for the observed effects on plant mineralization rates rather than a change in the proportion of particle sizes in each soil texture. Nevertheless, the results on texture controlling soil organic matter mineralization are contrasting in the literature, with results being dependent on soil moisture and other experimental conditions (Li et al., 2020; Li et al., 2022). Our experiment using artificial soils with precise composition and an early formation stage helped shed light on the effects of textures and particle size on organic matter decomposition.

**4.2 Interactions between clay and biochar in enhancing soil pH, and the consequences of early mineralization of plant residues**

The pronounced biochar effect on pH is likely due to its high acid buffering capacity. Biochar consists of both alkaline functional groups and mineral ash containing both base cations and secondary carbonates (Fidel et al., 2017), explaining why the biochar and clay + silt content have increased soil pH. The higher pH of the soil solution promotes the dissolution of $CO_2$ gas, thereby reducing the amount of gas released from soils. This can result in a measurement artifact of lower-than-actual mineralization rates in pH-enhanced soils (Ma et al., 2013). However, this effect was accounted for in our study, as

mineralization rates were corrected for amounts of $CO_2$ dissolved in solution as a function of solution volume and pH (Appelo
and Postma, 2005).
The liming effect of biochar is recognized in a variety of soils (Bolan et al., 2023), with an overall higher effect in
clay than in sandy soils (Ajayi and Rainer, 2017), which agrees with the findings of our experiment (Figure 3a). Nonetheless,
soil comparisons in natural conditions also vary in mineral composition, making the isolated textural effect less clear (Ajayi
and Rainer, 2017). In this sense, our artificial soil setup helps to clarify this effect by demonstrating a significant role of clay
and silt fractions in increasing the soil pH (Figure 3b) in soils with the same mineral composition and organic matter content.
Our results also showcase that this influence of clay and silt particles on soil pH is significantly boosted in biochar-amended
soils, as the slope of the linear correlation was three times higher in the biochar treatment than in the control (Figure 3b).
Direct sorption of $CO_2$ into biochar has also been reported alongside $N_2O$ adsorption. It can also be considered as a
mechanism through which biochar can reduce the presence of these gases in the atmosphere (Cornelissen et al., 2013). This
effect was probably counterbalanced by a higher clay + silt content in the soils, which may have protected the biochar surfaces
and thus prevented $CO_2$ sorption, justifying the relatively higher increases in C mineralization in the function of clay + silt in
biochar-amended soils (Figure 2). A higher soil pH has been shown to increase the mineralization of organic residues in soils
(Li et al., 2006, Khalil et al., 2005). This effect has been attributed to a higher microbial biomass and a more bacteria-dominated
microbial community in high-pH soils (Li et al., 2006, Laura 1976). In addition, a high pH causes stress for the microbial
community, resulting in lower carbon use efficiency, translating into lower assimilation for organic C and a higher $CO_2$ release
(Li et al., 2006). Some authors propose that the higher mineralization of organic matter may foster the supply of nutrients like
nitrogen and sulphur, further stimulating microbial activity (Barrow & Hartemink, 2023). Due to mineral surface availability,
a high clay + silt content may have levelled out the biochar effect. Sorption of cationic nutrients on those mineral surfaces may
have led to limited availability, reducing the $CO_2$ release in the mixtures with biochar and high clay + silt content (Barrow &
Hartemink, 2023). Therefore, we encourage further studies exploring the specific roles of microorganisms in artificial soil
setups to understand how they can influence these microcosms and how these results compare to natural conditions.
In conclusion, our study suggests that biochar has a higher capacity to promote reductions in the early mineralization
of clover residues in sandy soils. This potential diminishes as the clay and silt content increases in the soil. Here, we show that
the artificial soil setup using precise composition is a suitable tool for investigating the effects of factors that are difficult to
control in natural soils due to their heterogeneity. Our mineralization rates are comparable to other studies using artificial soils
(Vogel et al., 2014, Bucka et al., 2021), but can differ from experiments done in natural conditions (Gross et al., 2021).
Therefore, interpretations made here are mainly applied to understand the intrinsic relationships between soil particle size and
the biochar application on soil C persistence but should not be accounted for purposes of quantifying C emissions from
determined soil types or land uses.

## 221 4. Conclusions

222       We report the results of a screening study using an experimental setup for high-frequency short-term measurements.

We have observed significant effects of biochar content in reducing early mineralization of clover residues in artificial soils.
Soil particle size influences lowering C mineralization, but the magnitude of these changes is negligible compared to biochar
effects. Biochar has demonstrated the potential to reduce the early mineralization of plant residues, especially in sandy soils.
This effect is diminished with increased clay and silt content in the soil, suggesting that biochar may compensate for the lack
of clay in sandy soils by promoting lower mineralization of organic matter. Results on soil organic matter persistence and
carbon sequestration must be confirmed using longer-term experiments. However, this first set of results demonstrates the
power of using standardized multi-texture artificial soils to study biochar and organic matter interaction in soils. Here we
suggest that this artificial soil setup is a valuable platform for understanding mechanisms associated with biochar in soils.

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

**Acknowledgements**
The authors thank Mag. Engelbert Pürrer and Aspanger (Aspang, Austria) for providing the bentonite and illite used in the
preparation of the artificial soils.

**Author contribution**
TI, SW, FB, EF, and DR designed the experiment, carried out the analyses, and wrote the manuscript with the support of the
other authors.

**Competing interests**
The authors declare that they have no conflict of interest.

**Data availability**
The data that support the findings of this study are available from the corresponding author upon reasonable request.