# Peer review of "Biochar reduces early-stage mineralization rates of plant residues more in coarse than fine-texture soils — an artificial soil approach."

_EGUsphere, 2024_

## Referee Comment (RC1)

**General comments**

Soil texture is believed to have a strong impact on the response of biochar amendment on C sequestration, although little is known about the mechanisms involved. This short communication sheds light on potential mechanisms involved in the role of particle size and mineral composition on the early-stage decomposition of soil organic matter. The use of an artificial soil where particle size and mineral composition is fully controlled is, to my point of view, a relevant approach to focus on these specific mechanisms. Comparing the effect of biochar with soils from different textures is very interesting.

However, all parts could be improved since there are still some missing points and lack of precisions. In the results, it is not clear if the effect of biochar addition on the plant mineralization is due to the composition of biochar or to its particle size. Therefore, I think necessary to precise the size of biochar particles added and maybe to discuss this point.
To me, this study doesn't evidence a clear effect of the texture on plant respiration in the control soil (without biochar). It is written that there is a statistic effect, but (1) results from statistics made on very small samples (n=3) should be treated with caution, (2) I am not convinced by the choice of the statistic test and (3) the curve does not show a clear trend. Hence, the results and discussion should not emphasize too much on this small tendency and not consider it as a clear result.

In the discussion, only physical explanations are presented to explain the effect of pH on plant mineralization. Additional explanations should be proposed regarding the effect of pH changes on the microbial communities and on the nutrient availabilities from the plant for the microorganisms.
The discussion should include more comparison with other studies using artificial soils. Indeed, the results from studies using artificial soils can be very different from studies using field soils (e.g. Gross et al. 2021). Although the approach of using artificial soil is relevant, the study should more emphasize on its limits to extend these results to real field soils.

**Specific comments**

**Introduction**
You write that the texture influences C mineralization. It would be interesting to develop a bit more about the type of effects. You should also introduce the known effects of biochar addition and texture change on soil pH, since you will then show results about pH.
If other studies used artificial soils to explore the effects of texture on soil respiration, please cite some of them and tell a bit more what has already be done.
You can eventually add a sentence and/or a reference to explain why you chose the hypothesis that "biochar could reduce organic soil mineralization, especially in coarser textured soils". Did previous article already find results in this sense? Because for example, Gross et al. 2021 (who you cited) found that "SOC increases after biochar application were higher in medium to fine grain textured soils than in soils with coarse grain sizes" and Wang et al. 2017 (who you also cited) concluded that "biochar had minimal impacts on microbial communities in a coarser textured soil". As far as I understand, both studies seem to go in the opposite direction of your hypothesis.

**Material & methods**
Artificial soil preparation: since the texture plays a central role in this study, you could add the limit sizes of each quartz grains fraction (sand, silt, clay). For instance, what are the maximal and minimal size of the sand particles? Did the added goethite, illite and bentonite particles have a specific size?

For the biochar used: how long was the Norwegian Spruce pyrolyzed under 700 °C? Was there a specific size of the biochar particles? I think that you could add the %C of biochar in the main text (not just in supplementary material) because then you will present results of C.

Incubation: Did you regularly adjust the water quantity during all the incubation experiment?

You should add in this part that you also measured the soil pH ! Give a short description of the method for pH measurement and/or the norm that you used.

Statistics: You used the LSD statistical test. Do you mean the Fisher's Least Significant Difference test? If yes, please precise it. For only 3 replicates, I would have rather used a non-parametric test because you cannot easily prove the normal distribution… Anyway, if you decided to use parametric tests, why did you not simply use the student t-test for each modality individually? Because then you always present each modality individually.

**Results**
*3.1 Early plant mineralization affected by particle size and biochar application*
Figure 1: I understand that you made 3 replicates for each modality. However, I see only 1 result per modality and time on your figure. Are the results means of the three replicates? It would be nice to have an idea of the variability among one modality on the graph.

Line 78: "Control samples' clay and silt content generally decreased C mineralization" → to me, it is not clearly shown by the figure 2. What I understand from the figure 2 is that the Clay+Silt content is positively correlated to the cumulative respiration in the modalities with biochar added, but I don't see any evident negative correlation with the control modalities (you could also add the equation + $R^2$ of the control modalities in figure 2).
You write that the texture has a significant impact. If yes, please give the p-value for both control and biochar modalities.

A thing that is interesting that you could tell more about, is the increase of cumulative respiration from the control in the first 33 hours of incubation (figure 1). The figure 1 clearly shows that the slope of the loamy sand control is lower than all the other soils. It would be interesting to compare this slope between all soils and emphasize this point. For that, a possibility could be to make a single graph with all results from cumulative respiration (if necessary, why not trying the logarithmic scale for the y-axis).

*3.2 Soil pH affected by particle size and biochar application*
Is there any effect of the texture on the soil pH? Why didn't you make statistical analysis about the texture effect, as for what you did in the 3.1?

Do you have significant differences between the modalities with biochar? For example, if the biochar modality with loamy sand significantly lower than the biochar with silty clay? If not, you can at least stress the tendency.

**Discussion**
Lines 97, 98: "The significant interaction … plant residues depends on soil texture" → you can write that this confirms many other studies and cite some of them (for ex. The ones you cited in the introduction).

Lines 100, 101: "biochar had a higher impact on reducing early C-mineralization from clover residues 101 in sandier textures than in clay-rich soils" → is it coherent with other studies, and especially with the studies using artificial soils? Please compare with what is known and if it is different from most studies, please emphasize on it.

Lines 105-108: do you mean physicochemical protections through organo-mineral complexation? If yes, it would be especially promoted by the fact that you used ball-milled powder of organic matter. Hence, the surface of the organic matter is maximal.

Lines 110-112: "The biochar effect is probably … physicochemical protection of OM" → Did other studies find that? Because as far as I know, the clay content is generally positively correlated with SOC content due to more organo-mineral complexes formations. Do you mean that clay surface competes with biochar surface for organo-mineral complexation?

Line 117: "Figure 1b" → Do you mean figure 2?

Lines 117-118: "Every mg of silt and clay size particles … clover residue mineralization" → I don't see where it is clearly shown in your results…

Lines 120-121: "Our results suggest that the increase of clay … early-stage C mineralization of crop residues" → repetition + idem comment as for lines 117-118

Lines 128-129: "The higher pH of the soil … gas released from soils" → did previous studies found that? If yes, please cite them.

Very interesting propositions to explain how the change of pH may affect microbial respiration. But your explanations only consider the physical aspects of soil properties (CO2 dissolution, CO2 sorption on biochar's surface). Very important is also the effect on microbial communities! How do they react to a change of pH? Explanations could also relate to the availability of nutrients from the organic matter for the microorganisms…

You present the effect of biochar addition on the C-mineralization rate. But you also write that the texture had a significant impact. If you decide to speak about the effect of the texture, it would be logical to tell a bit more about the effect of the texture. For example, did the control present significant changes of C-mineralization depending on the texture? I think it is important to precise here. Otherwise, you don't know if the effect of biochar addition is due to its composition or to its particle size…

Gross et al. 2021 concluded that treatments conducted in greenhouses and laboratories can induce significantly higher responses in SOC sequestration potential when compared to treatments observed on a field scale. → Need to discuss the representativity of the artificial soil, as compared to soil in field.

***Conclusions***
Line 138: "significant effects of biochar" → "significant effects of biochar **content**"
Line 138: "soil texture in reducing early mineralization" → please precise which increase of which texture reduces the early mineralization. + I am not convinced by your conclusion that the silt+clay content is negatively correlated with soil respiration.

Lien 141: "biochar may compensate for the lack of clay in sandy soils" → is it due to the particle size of the biochar?

Line 145: "Terra Preta soils » → the link is really not clear with your study, since you worked with an artificial soil and Terra Preta are real soils. Please precise the link.

**Technical corrections**
compact listing of purely technical corrections at the very end (typing errors, etc.).

line 53: "…according to **(**Pronk et al. 2012)" → "…according to Pronk et al. **(**2012)"
line 59: "CO2" → "$CO_2$"
Line 103: remove the dot after (Obia et al. 2016)
Line 111: "diminished with" → "reduced by"
Line 132: "can also be considered a mechanism" → "can also be considered **as** a mechanism"

---

## Author Comment (AC1)

**Biochar reduces early-stage mineralization rates of plant residues more in coarse than fine-texture soils – an artificial soil approach**

Thiago M. Inagaki, Simon Weldon, Franziska B. Bucka, Eva Farkas, and Daniel P. Rasse
**Status**: Final response (EGUsphere)

*We thank the editor and the reviewers for taking the time to evaluate the manuscript and for considering our manuscript suitable for publication. We also thank the reviewer for providing thoughtful comments and constructive feedback that helped us to improve the manuscript. We have carefully revised our manuscript to address the issues raised in the reviewer's comments. We have written a point-by-point response to the comments (indicated in blue and italics) and marked our changes in the manuscript.*

**Response to Reviewer Comments**
RC1: 'Comment on egusphere-2024-1143', Marie-Liesse Aubertin, 11 Jun 2024

General comments (specific comments and technical corrections are in the pdf file):
Soil texture is believed to have a strong impact on the response of biochar amendment on C sequestration, although little is known about the mechanisms involved. This short communication sheds light on potential mechanisms involved in the role of particle size and mineral composition on the early-stage decomposition of soil organic matter. The use of an artificial soil where particle size and mineral composition is fully controlled is, to my point of view, a relevant approach to focus on these specific mechanisms. Comparing the effect of biochar with soils from different textures is very interesting.
*Author's response: Thank you for your kind and constructive comments. We appreciate your acknowledgment of this study's innovations and suggestions for improvement. We did our best to address all the reviewers' comments. We believe that the manuscript is now clearer and more informative.*

However, all parts could be improved since there are still some missing points and lack of precisions. In the results, it is not clear if the effect of biochar addition on the plant mineralization is due to the composition of biochar or to its particle size. Therefore, I think necessary to precise the size of biochar particles added and maybe to discuss this point.
*Author's response: Thank you for your observation. The biochar added in the experiment was between 2 mm and 0.063 mm. Therefore, we believe the changes in mineralization patterns are mainly due to the composition of the biochar rather than changes in particle size in the different textures. We have now added in the Material and Methods "Section 2.2 Incubation Experiment and Treatments" the details about the biochar size added. We have also added to the Discussion Section "4.1 Interactions of clay + silt content and biochar application in reducing early mineralization of plant residues" (second paragraph) an explanation regarding the effects of biochar particle size vs. composition. We believe these two distinct effects are more clearly differentiated in the manuscript.*

To me, this study doesn't evidence a clear effect of the texture on plant respiration in the control soil (without biochar). It is written that there is a statistic effect, but (1) results from statistics made on very small samples (n=3) should be treated with caution, (2) I am not convinced by the choice of the statistic test and (3) the curve does not show a clear trend. Hence, the results and discussion should not emphasize too much on this small tendency and not consider it as a clear result.
*Authors' response: Thank you for your observation. Indeed, we agree that despite the significant correlations, the influence of clay and silt particles is quite minimal in our experiment, given the low slope of the curve. Therefore, as suggested, we have changed the discussion of the manuscript by minimizing the interpretations of the textural influence on plant mineralization (Discussion section 4.1, third paragraph). More details are given in the response to the specific comments.*

In the discussion, only physical explanations are presented to explain the effect of pH on plant mineralization. Additional explanations should be proposed regarding the effect of pH changes on the microbial communities and on the nutrient availabilities from the plant for the microorganisms.
*Author's response: Thanks for the suggestion. We have elaborated our discussion of the pH effect and added a section on the biological effects of pH changes on OC decomposability by soil, the soil microbial community, and the potential availability of nutrients. In addition, we supported our discussion of the biological effects of a changed soil pH by adding more literature references (Lines 166 – 171).*

The discussion should include more comparison with other studies using artificial soils. Indeed, the results from studies using artificial soils can be very different from studies using field soils (e.g. Gross et al. 2021). Although the approach of using artificial soil is relevant, the study should more emphasize on its limits to extend these results to real field soils.

*Authors' response: Thank you for this critical observation. Indeed, the interpretations made here mainly focus on understanding the intrinsic relationships between soil particle size and biochar and how they affect C persistence. Nonetheless, such mineralization rates should not be used, e.g., quantifying C emissions from determined soil types or land uses, since this is not the purpose of this study. We now emphasize that in the meta-analysis conducted by Gross et al. 2021, the trend opposite to our findings was exclusively for field experiments, while the authors found the same trend for non-field experiments. These differences are now more comprehensively discussed in the discussion section (Lines 144-157) and the introduction (Lines 26 – 33). We also emphasize better now to what situations this study can be applied in the discussion section (Lines 139 – 149) (non-field setup) so we do not overinterpret the outcomes of this experiment. We have added more references in the discussion section by comparing our findings with other studies in natural conditions and emphasizing how the use of artificial soils can help to go beyond the intrinsic relationships between specific factors, such as biochar and soil texture, the topic of our study. We have also clarified this aspect in Section 4.2, "Interactions between clay and biochar in enhancing soil pH, and the consequences of early mineralization of plant residues" (last paragraph), by indicating other experiments done in artificial soils and emphasizing the scope of this experiment. We believe that the central message of this study is now more precise, and the limitations are more emphasized. We also mention more specific factors in this question in the answers to specific comments.*

**Specific comments**

**Introduction**

You write that the texture influences C mineralization. It would be interesting to develop a bit more about the type of effects. You should also introduce the known effects of biochar addition and texture change on soil pH, since you will then show results about pH.
If other studies used artificial soils to explore the effects of texture on soil respiration, please cite some of them and tell a bit more what has already be done.
*Author's response: Thank you for your suggestion. We have now added complementary information in the introduction, discussing the effects of biochar in changing the soil pH under contrasting textures (Lines 32 – 37) and providing references to other experiments using artificial soils. We then made a clearer connection between the potential of artificial soil setups to investigate the research gaps in biochar studies. We believe now that our introduction better presents the issues we are investigating and the research gaps in biochar studies.*

You can eventually add a sentence and/or a reference to explain why you chose the hypothesis that "biochar could reduce organic soil mineralization, especially in coarser textured soils". Did previous article already find results in this sense? Because for example, Gross et al. 2021 (who you cited) found that "SOC increases after biochar application were higher in medium to fine grain textured soils than in soils with coarse grain sizes" and Wang et al. 2017 (who you also cited) concluded that "biochar had minimal impacts on microbial communities in a coarser textured soil". As far as I understand, both studies seem to go in the opposite direction of your hypothesis.
*Author's response: Thank you for your observation. We have clarified the reasons behind the contrasting results found in the literature and how our work helps us understand this question better. The reasons for these discrepancies seem to depend mainly on the initial condition of the evaluated soils. For example, in the incubation setup comparing contrasting textures in Wang et al. (2017), the unamended samples had no difference in water-stable aggregates despite their contrasting texture, and the C content was higher in the sandy soil. This experiment resulted in higher impacts of biochar application on clay soils' structure and C storage. On the other hand, in our experiment, the unamended samples with higher clay content had a significantly higher water-hold capacity (Supplementary Figure 1), indicating an overall better structure than the sandy soils, and the organic matter added to them was the same for all soils, which resulted in a better performance of biochar in more sandy textures. Therefore, the impacts of biochar on reducing C mineralization in our experiment suggest that the amendment can compensate for the lack of soil structural quality in promoting early C persistence in soils. Likewise, Gross et al. (2021) pointed out the low initial soil C content of sandier textures as a reason for observing higher SOC sequestration potential in non-field experiments. The contrast with field experiments can also be due to factors that happen only in field trial setups, such as biochar leaching in sandy soils, which are not observed under pot or incubation setups. Therefore, our interpretations are focused more on the intrinsic relationships between soil texture and biochar and their impacts on C mineralization in a controlled environment. However, climatic factors and long-term effects need to be confirmed through long-term field trials. Nonetheless, since most of the works evaluating the performance of biochar in different soil textures vary in other factors such as C content, mineralogy, and structural properties, our work offers important insights to understanding these interactions of biochar with*

*soil particle size in C mineralization. We have now included this discussion in lines 144-165 and the introduction (Lines 26 – 32).*

**Material & methods**

Artificial soil preparation: since the texture plays a central role in this study, you could add the limit sizes of each quartz grains fraction (sand, silt, clay). For instance, what are the maximal and minimal size of the sand particles? Did the added goethite, illite and bentonite particles have a specific size?

*Author's response: We have now added the size of each grain particle in Table 1. They were chosen specifically to represent the particle grain size of sand, silt, and clay.*

For the biochar used: how long was the Norwegian Spruce pyrolyzed under 700 °C? Was there a specific size of the biochar particles? I think that you could add the %C of biochar in the main text (not just in supplementary material) because then you will present results of C.

*Author's response: The biochar was produced with a holding time of 7 minutes at 700°C, which is now included in the article, Line 73. The biochar particles added in the experiment were sieved between 2 mm and 0.063 mm, which we have now included in the Material and Methods Lines 74-75. The biochar C content is now specified in Line 74.*

Incubation: Did you regularly adjust the water quantity during all the incubation experiment?

*Author's response: We have monitored the weight of the samples during the incubation period to check the necessity of adjusting the water content. Since this was a relatively short incubation conducted in a closed environment, the water loss was minimal, and we did not need to refill the flasks to maintain the same moisture content. We have now added this observation to the Material and Methods section 2.3.*

You should add in this part that you also measured the soil pH ! Give a short description of the method for pH measurement and/or the norm that you used.

*Author's response: Thank you. We forgot to mention the pH analysis. We have now added the details on Line 87.*

Statistics: You used the LSD statistical test. Do you mean the Fisher's Least Significant Difference test? If yes, please precise it. For only 3 replicates, I would have rather used a non-parametric test because you cannot easily prove the normal distribution… Anyway, if you decided to use parametric tests, why did you not simply use the student t-test for each modality individually? Because then you always present each modality individually.

*Author's response: Thank you for your observation. Yes, the LSD test used was the Fisher's Least Significant Difference test. We recognize the lack of clarity in the statistical analysis description and acknowledge your concern about the small sample size. We have now provided a more detailed description of our statistical approach in the Material and Methods section by adding a new sub-section, "2.4 Statistical Analysis". We performed a Shapiro-Wilk test in our dataset, indicating a normal distribution of the cumulative mineralization and pH data. Next, we conducted a two-way ANOVA, which showed a significant interaction between biochar application and soil textures. We then conducted a posthoc test using Fisher's Least Significant Difference test (LSD) to compare biochar and control treatments within each soil textural group. To address the reviewer's concerns, we have reanalyzed our data using the t-test instead of the LSD and a non-parametrical Kruskal-Wallis test, and the outcomes were identical. Therefore, we opted to maintain our original approach to analyze the data, but now with a more detailed description in the manuscript. Given the consistent results across different statistical methods and the evidence supporting the normality assumption, we believe that retaining the ANOVA + LSD test is justified.*

**Results**

*3.1 Early plant mineralization affected by particle size and biochar application*

Figure 1: I understand that you made 3 replicates for each modality. However, I see only 1 result per modality and time on your figure. Are the results means of the three replicates? It would be nice to have an idea of the variability among one modality on the graph.

*Author's response: The points in Figure 1 are the average of three replicates, which we have now specified in the Figure. Since this is an artificial soil experiment, the variation is minimal; the standard deviation bars are smaller than the points themselves, making them invisible.*

Line 78: "Control samples' clay and silt content generally decreased C mineralization" → to me, it is not clearly shown by the figure 2. What I understand from the figure 2 is that the Clay+Silt content is positively correlated to the cumulative respiration in the modalities with biochar added, but I don't see any evident negative correlation with the control modalities (you could also add the equation + R2 of the control modalities in figure 2).
You write that the texture has a significant impact. If yes, please give the p-value for both control and biochar modalities.

*Author's response: Thank you for your observation. Indeed, we forgot to insert the equation and p-value for the control treatment, which was also significant. We have now corrected the Figure. The increased chances for physicochemical protection with the increase of clay and silt particles justifies the decrease in mineralization with the rise in silt and clay in control samples. However, since this is an experiment with artificial soils with precise composition, statistical tests tend to find more significant results in correlations due to the uniformity of the data. After re-interpreting the data, we recognize that despite the significant correlations, the changes in mineralization caused by soil texture alone are negligible compared to the ones caused by biochar, given the low slope values in the linear correlations. Therefore, we now emphasize these aspects in the discussion section (Lines 167 – 172).*

A thing that is interesting that you could tell more about, is the increase of cumulative respiration from the control in the first 33 hours of incubation (figure 1). The figure 1 clearly shows that the slope of the loamy sand control is lower than all the other soils. It would be interesting to compare this slope between all soils and emphasize this point. For that, a possibility could be to make a single graph with all results from cumulative respiration (if necessary, why not trying the logarithmic scale for the y-axis).

*Author's response: Thank you for your suggestions. As you mentioned, we indeed observed a difference in mineralization patterns during the first hours of incubation. However, since our automatized GC cannot measure all the samples simultaneously, we recognize that the difference in initial patterns may be caused by a difference in time measurement (by measuring the samples one after the other). Therefore, we decided to limit our interpretations to the cumulative respiration at the end of the incubation period.*

*3.2 Soil pH affected by particle size and biochar application*
Is there any effect of the texture on the soil pH? Why didn't you make statistical analysis about the texture effect, as for what you did in the 3.1?

*Author's response: The texture positively increased the soil pH, as shown in Figure 3 (b). Since soil texture is strictly controlled, as it is an artificial soil setup, we opted to correlate the variables through linear regressions instead of comparing averages of different categories.*

Do you have significant differences between the modalities with biochar? For example, if the biochar modality with loamy sand significantly lower than the biochar with silty clay? If not, you can at least stress the tendency.
*Author's response: Yes, we have now discussed these trends more, as discussed in the previous answer on Lines 194-197*

**Discussion**

Lines 97, 98: "The significant interaction … plant residues depends on soil texture" → you can write that this confirms many other studies and cite some of them (for ex. The ones you cited in the introduction).
*Author's response: Thank you. We have now added more references to this part.*

Lines 100, 101: "biochar had a higher impact on reducing early C-mineralization from clover residues in sandier textures than in clay-rich soils" → is it coherent with other studies, and especially with the studies using artificial soils? Please compare with what is known and if it is different from most studies, please emphasize on it.
*Author's response: Thank you for your observation. We have added further references in our discussion section, showcasing how our results using artificial soils are comparable with other experiments using natural soils. However, to our knowledge, no other studies have compared different soil textures using biochar in artificial soils. This is one of the main innovations that our study brings. Since most of the works evaluating the performance of biochar in different soil textures have variations in other factors such as C content, mineralogy, and structural properties, the conclusions regarding the interactions of biochar with soil particle size in C dynamics are hampered. Therefore, our setup using artificial soils with identical C content and mineralogy, varying only in texture, helps to shed light on the mechanisms behind these interactions.*

Lines 105-108: do you mean physicochemical protections through organo-mineral complexation? If yes, it would be especially promoted by the fact that you used ball-milled powder of organic matter. Hence, the surface of the organic matter is maximal.

*Author's response: Since the inorganic ingredients used to compose these soils are C-free, we used ball-milled organic matter to enhance soil development. Nonetheless, the organic matter added was the same for all treatments, standardizing their effect. We have added this information to the Material and Methods Section Lines 68-69.*

Lines 110-112: "The biochar effect is probably … physicochemical protection of OM" → Did other studies find that? Because as far as I know, the clay content is generally positively correlated with SOC content due to more organo-mineral complexes formations. Do you mean that clay surface competes with biochar surface for organo-mineral complexation?

*Author's response: Yes, the role of such physicochemical interactions is recognized as a major reason for the OM protection promoted by biochar. Biochar can promote soil physical improvements and the direct sorption of several components, facilitating the overall higher persistence of C in soil. Likewise, soil clay and silt fractions promote similar physicochemical protection of organic matter. Our results suggest that the increase of clay and silt content may overrule the effects of biochar in reducing this early mineralization of plant residues, which justifies the higher biochar effects on sandy soils. Such reduction in biochar's effect with clay increase could potentially be due to competition on sorption areas or blockage of pores in the biochar structure by clay particles. We have now added further information about it in Lines 196-200.*

Line 117: "Figure 1b" → Do you mean figure 2?
*Author's response: Yes, thank you. We have now corrected the reference*

Lines 117-118: "Every mg of silt and clay size particles … clover residue mineralization" → I don't see where it is clearly shown in your results…
*Author's response: This is observed in the slope of the linear correlation equation in the Figure. We have now specified it in the text (Lines 125-126)*

Lines 120-121: "Our results suggest that the increase of clay … early-stage C mineralization of crop residues" → repetition + idem comment as for lines 117-118
*Author's response: We have now reformulated this discussion to address the issue.*

Lines 128-129: "The higher pH of the soil … gas released from soils" → did previous studies found that? If yes, please cite them.
*Author's response: The reference was mentioned in the following phrase. We have now merged the phrases to address the issue (Lines 191-192)*

Very interesting propositions to explain how the change of pH may affect microbial respiration. But your explanations only consider the physical aspects of soil properties (CO2 dissolution, CO2 sorption on biochar's surface). Very important is also the effect on microbial communities! How do they react to a change of pH? Explanations could also relate to the availability of nutrients from the organic matter for the microorganisms…
*Author's response: Thank you for your suggestions. We have now added further discussions in lines 177-190. See also our response in the general comments*

You present the effect of biochar addition on the C-mineralization rate. But you also write that the texture had a significant impact. If you decide to speak about the effect of the texture, it would be logical to tell a bit more about the effect of the texture. For example, did the control present significant changes of C-mineralization depending on the texture? I think it is important to precise here. Otherwise, you don't know if the effect of biochar addition is due to its composition or to its particle size…
*Author's response: Since we have now improved the discussion regarding the textural effects of the impacts of biochar on C mineralization, we believe this issue is more evident in the revised version of our manuscript. As mentioned, the effects of texture alone on C mineralization were minimal compared to the impact of biochar application. Also, the biochar particle size added in the experiment was between 0.063 and 2 mm (now stated in the Material and Methods), which implies that biochar size was likely not a major influencer in the changes observed in C mineralization.*

Gross et al. 2021 concluded that treatments conducted in greenhouses and laboratories can induce significantly higher responses in SOC sequestration potential when compared to treatments observed on a field scale. → Need to discuss the representativity of the artificial soil, as compared to soil in field.

*Author's response: We have discussed it more thoroughly in the introduction and discussion section, as emphasized in previous responses. Our artificial soil setup is more comparable with non-field experiments. Factors such as the leaching of biochar in sandy soils are not accounted for in this kind of experiment, but they can potentially happen in field conditions. These aspects are now further discussed in Lines 145-158*

**Conclusions**

Line 138: "significant effects of biochar" → "significant effects of biochar content"
*Author's response: We have now provided the change in the phrase.*

Line 138: "soil texture in reducing early mineralization" → please precise which increase of which texture reduces the early mineralization. + I am not convinced by your conclusion that the silt+clay content is negatively correlated with soil respiration.
*Author's response: We have now modified this conclusion based on the new interpretations about the influence of clay and silt particles on C mineralization.*

Lien 141: "biochar may compensate for the lack of clay in sandy soils" → is it due to the particle size of the biochar?
*Author's response: As mentioned in the previous answers, this issue is better clarified throughout the manuscript.*

Line 145: "Terra Preta soils » → the link is really not clear with your study, since you worked with an artificial soil and Terra Preta are real soils. Please precise the link.
*Author's response: We have now replaced Terra Preta soils with Biochar.*

**Technical corrections**
compact listing of purely technical corrections at the very end (typing errors, etc.).

line 53: "…according to (Pronk et al. 2012)" → "…according to Pronk et al. (2012)"
*Author's response: We corrected the term.*

line 59: "CO2" → "CO2"
*Author's response: We corrected the term.*

Line 103: remove the dot after (Obia et al. 2016)
*Author's response: We corrected the term.*

Line 111: "diminished with" → "reduced by"
*Author's response: We corrected the term.*

Line 132: "can also be considered a mechanism" → "can also be considered as a mechanism"
*Author's response: We corrected the term.*

**Reviewer 2**
The authors present a study investigating the effect of biochar on carbon mineralization in soils of different texture. Although technically sound and well presented, in my opinion, the study as written is insufficient to conclusively corroborate the assertions and conclusions made by the authors. I strongly feel the authors could rewrite and present a stronger case for their study with more experimentation (e.g. utilizing more biochars from different feedstocks and comparing or using the same feedstock but varying temperature of pyrolysis or investigating effect of pH change alone without biochar. Increased  pH or liming alone has been well documented to impact C mineralization with increased pH encouraging aggregation of clays and thus increased protection of carbon within the aggregates and binding to Ca2+ and clay surfaces.
*Author's response: We acknowledge the reviewer's observation regarding the scope of analyses and treatments but believe the novelty of our findings and their potential to promote further investigation in biochar research*

*outweigh the need for a more extensive initial study, making a Short Communication the ideal format for presenting these concise, high-impact results. In the revised manuscript, we have made significant improvements to better illustrate this study's main innovations and how they help to understand questions regarding using biochar in soils with different textures. Despite the variety of studies evaluating the effects of biochar on C storage and liming, much information on how soil texture affects these impacts remains unknown, given that many conflicting results exist in the literature. Besides the lack of studies under a broader range of conditions, a significant reason for these uncertainties is the heterogeneous nature of soils, which often hampers interpretations regarding the exact factors responsible for the observed changes. In this sense, we showcase in this short experiment that using artificial soils to investigate the intrinsic relationships between biochar and soil particle size using soils with pre-made exact compositions can be an excellent and underutilized tool. To our knowledge, no other studies have systematically investigated using biochar under a gradient of soil textures with the same mineralogy and C amount. Also, no other studies have used artificial soils to study these interactions. We understand the desire for more comprehensive analyses. Nonetheless, we argue that even incremental advancements, such as those presented in our manuscript, play a crucial role in the progression of biochar research. Publishing these findings as a short communication allows us to promptly share valuable information with the scientific community, fostering a more dynamic exchange of ideas and accelerating overall progress.*